# Direct-Read Fluorescence-Based Measurements of Bioaerosol Exposure in Home Healthcare

**DOI:** 10.3390/ijerph19063613

**Published:** 2022-03-18

**Authors:** Vishal D. Nathu, Jurate Virkutyte, Marepalli B. Rao, Marina Nieto-Caballero, Mark Hernandez, Tiina Reponen

**Affiliations:** 1Department of Environmental & Public Health Sciences, College of Medicine, University of Cincinnati, Cincinnati, OH 45267-0056, USA; vishalnathu18@hotmail.com (V.D.N.); virkutyte.jurate@gmail.com (J.V.); raomb@ucmail.uc.edu (M.B.R.); 2Department of Environmental Engineering, College of Engineering & Applied Science, University of Colorado Boulder, Boulder, CO 80309-0428, USA; marina.nietocaballero@colorado.edu (M.N.-C.); mark.hernandez@colorado.edu (M.H.)

**Keywords:** bioaerosol, home healthcare, residential, occupational, fluorescence, activities, direct read

## Abstract

Home healthcare workers (HHCWs) are subjected to variable working environments which increase their risk of being exposed to numerous occupational hazards. One of the potential occupational hazards within the industry includes exposure to bioaerosols. This study aimed to characterize concentrations of three types of bioaerosols utilizing a novel fluorescence-based direct-reading instrument during seven activities that HHCWs typically encounter in patients’ homes. Bioaerosols were measured in an indoor residence throughout all seasons in Cincinnati, OH, USA. A fluorescence-based direct-reading instrument (InstaScope, DetectionTek, Boulder, CO, USA) was utilized for all data collection. Total particle counts and concentrations for each particle type, including fluorescent and non-fluorescent particles, were utilized to form the response variable, a normalized concentration calculated as a ratio of concentration during activity to the background concentration. Walking experiments produced a median concentration ratio of 52.45 and 2.77 for pollen and fungi, respectively. Fungi and bacteria produced the highest and lowest median concentration ratios of 17.81 and 1.90 for showering, respectively. Lastly, our current study showed that sleeping activity did not increase bioaerosol concentrations. We further conclude that utilizing direct-reading methods may save time and effort in bioaerosol-exposure assessment.

## 1. Introduction

Home healthcare is one of the fastest-growing industries, comprising of different types of laborers such as physicians, nurses, aides, and other skilled workers. Home healthcare workers (HHCWs) are subjected to variable working environments which increase their risk of being exposed to numerous occupational hazards [1]. Exposures to occupational hazards are similar to HHCWs in the essence that they work in indoor home environments; however, the uniqueness of each home causes distinctive types of exposures [1]. There are minimal data on exposures to hazards in the residential occupational environment. Furthermore, the information related to health effects from occupational exposures to HHCWs is also limited; however, there is sufficient evidence that home healthcare is not free from occupational hazards [2]. Occupational hygiene practices such as identifying risks to HHCWs are challenging due to unpredictable home care environments and privacy requirements. In addition, patient homes differ from typical healthcare settings, making them unregulated and potentially unsafe for HHWCs.

The growing home healthcare industry employs two distinct types of labor: home care workers and home healthcare providers. The Institute for Healthcare Improvement defines “home care workers” as individuals who aid in residences with activities of daily living and “home healthcare workers” as skilled workers providing medical care to patients [1,3]. Other unlicensed workers within the home healthcare industry include patient care aides who support patients with activities such as showering, feeding, and cleaning. Licensed and unlicensed workers within the industry have similar occupational exposures when working within unpredictable and uncontrolled residential indoor environments; however, differences are found based on tasks performed [2,4]. These tasks are quickly increasing as population demographics within the United States depend on home healthcare due to aging, chronic diseases, and the inability to perform activities of daily living [5]. According to the Bureau of Labor Statistics, the number of home care aides aiding with activities of daily living is expected to increase nearly 40 percent by 2026 to over 2.7 million jobs, thus causing workers to be exposed to numerous types of occupational hazards [6].

One of the potential occupational hazards within the industry includes exposure to bioaerosols. Bioaerosols are particles of biological origin in the air produced from animals, plants, bacteria, fungi, and humans [7,8,9,10]. The small size and light weight of bioaerosols cause them to be readily transported throughout the air, allowing them to remain airborne for an extended time in indoor air environments [11]. Indoor bioaerosols can impact human health and may cause adverse health effects [12]. In addition, the gaps in knowledge are substantial when referring to indoor bioaerosols, which are relevant due to the vast majority of the time people spend indoors [13,14,15]. Investigators have conducted experiments on bioaerosol sources, exposure–response relationships, transmission, and sampling/detection methods [7,11,16,17,18]. Residential areas where bioaerosol concentrations may be elevated in indoor air include bathrooms and kitchens [11,19,20,21].

Indoor bioaerosol sources are dependent on various factors such as human occupancy, occupant behavior, and extrinsic activities such as vacuuming and cooking, as well as intrinsic activities such as expiratory flow and shedding from the skin and clothing [11,22,23,24,25,26]. The presence of humans and types of activities can also impact both the type and concentration of bioaerosols in indoor environments [14,23,24,27,28,29]. Human interactions are also able to influence levels of airborne bioaerosol from skin shedding, household materials such as upholstery, flooring, mattresses, and performing activities such as cleaning, vacuuming, and showering [14,21,23,26,30,31,32,33,34,35,36,37]. Dust in residential environments may also contain a wide range of biological content, such as molds. When performing activities such as vacuuming, components of the vacuum may act as an agent in the emission and aerosolization of bioaerosols, or reaerosolization from the surface being vacuumed [15,38]. Bioaerosols are also present in the bedroom, where individuals spend more than a third of their lives [27,39,40]. Home healthcare workers attend to patients who may be bound to their bedrooms during care. Occupational exposure to bioaerosols in home healthcare is under characterized. The need to fill such a data gap is crucial in protecting workers from hazards, especially because they are subjected to performing activities of daily living within the residential indoor environment.

Conventional bioaerosol sampling methods include obtaining samples via filtration, impaction, and impingement. Oftentimes, obtaining samples such as these requires time and laboratory effort for analyses. Furthermore, each method is specific for the assessment of only one type of bioaerosol particles (e.g., pollen, fungi, or bacteria). Although traditional occupational exposure monitoring can capture exposures to bioaerosols, real-time monitoring and direct-reading capability are starting to become of interest. Thus, a novel approach is needed to characterize the real-time exposure to various types of bioaerosols simultaneously. There are limited publications on real-time monitoring of indoor bioaerosols [27,41,42,43] in the scientific literature. Such ability to measure in real time and obtain larger datasets would allow agencies to develop guidelines for protecting home healthcare’s growing industry and protecting HHCWs.

The goal of this study was to characterize concentrations and types of bioaerosols utilizing a novel fluorescence-based direct-reading instrument during common activities that HHCWs encounter in patients’ homes. We investigated the effect of seven different activities on concentrations of bacteria, fungi, pollen, and fluorescent and non-fluorescent particles. We also report the real-time bioaerosol generation for two statistically significant activities and show distinct peaks during activities that correspond to the possible disturbance of particles. 

## 2. Materials and Methods

Bioaerosols were measured in an indoor residence during 2018–2019 throughout all seasons in Cincinnati, OH, USA. The ranch-style residence was built in 2003 and included three bedrooms and two bathrooms (Figure 1). The tested bedroom, bathroom, kitchen/dining room, and living room were approximately 3, 20, 40, and 24 m^2^, respectively. The outer material of the home was standard brick, while the internal residence included areas with carpeting and hardwood flooring. The residence was supplied with a mechanical heating, ventilation, and air conditioning (HVAC) split system (2.5-ton 14 SEER R-410A) equipped with standard pleated air filters with MERV-8 ratings [44]. The HVAC system was serviced yearly in early spring (March/April). Furthermore, the home had natural ventilation (through windows) in the rooms. Windows were only opened within the specific room during measurements (i.e., bedsheet changing included windows open within the bedroom, cooking included windows open within the kitchen, etc.). The bathroom within the home did not have windows. The typical inside temperature was 72 °F (HVAC mode) and 40 to 80 °F when the windows were open. A total of three human occupants were living in the home, along with one pet cat. There appeared to be no visible water damage or other biological material such as plants in any of the living spaces.

A fluorescence-based direct-reading instrument (InstaScope, DetectionTek, Boulder, CO, USA) was utilized for all data collection. The operational principles of the InstaScope included a mobile version of the Wideband Integrated Bioaerosol Sensor (WIBS) (DetectionTek, Boulder, CO, USA) and are described elsewhere [45,46,47,48,49]. In brief, the InstaScope was equipped with an optical reference library of biological signatures to characterize airborne bioaerosols [42]. The manufacturer performed calibration of the InstaScope before the study [49].

Particles were sampled through an inlet wand at 0.835 L/min and entered a chamber equipped with a continuous diode laser to produce scattered light, which two Xenon flashlamps then detected. Upon detection, if the particle emitted fluorescence due to excitation, two photomultiplier tubes designed to detect such emissions would then log total counts, particle size, and fluorescence properties. After a monitoring event, the raw WIBS data were extracted from the instrument via external hard drives.

The particles were characterized as one of seven types first introduced by Perring et al. [46] which considered three fluorescence bandwidths individually and in all possible combinations. The WIBS instrument utilizes two excitation wavelengths, 280 and 370 nm. Fluorescence emitted by particles is recorded by three channels (Channel A, Channel B, and Channel C). The notations TYPE A280 = (310 − 400), TYPE B280 = (420 − 650), and TYPE C370 = (420 − 650) were utilized. Each subscripted integer denotes excitation wavelengths, while the parentheses indicate the emission bandwidths observed [45]. Any particle could have signaled above the fluorescence threshold within these channels, leading to seven (A, B, C, AB, AC, BC, and ABC) likely combinations of fluorescence signal [46]. Bacterial, fungal, and pollen particle classification based on fluorescence and channel type are described elsewhere [45].

To investigate the effect of activities of daily living on bioaerosol generation, a total of four background measurements (averaging 52 min and ranging from 49 min to 53 min) were taken prior to the investigated activities of daily living. Background measurements were performed as far as possible from the sampling rooms within the home. The corridor closest to the main entrance was chosen for background measurements (Figure 1).

We then investigated seven activities (bedsheet changing, cooking, dusting, showering, sleeping, vacuuming, and walking) which HHCWs perform for patients. Each activity was replicated five times, except for sleeping (*n* = 4). The activities were monitored for an average of 65 min (ranging from 28 to 224 min). We simulated actual real-life working conditions of HHCWs but did not study real HHCWs; therefore, the number of people in these areas ranged from one to three people. Additional study participants were not recruited to perform the activities of daily living. The dusting activity was conducted in the bedroom, living room, and kitchen. The walking activity was simulated in all of the rooms. Vacuuming was performed in the bedroom and living room (Figure 1). During all activities, the InstaScope sampling wand was capable of dynamic movement via wheels and followed the activities approximately 70 cm in height and 70 cm from the point-of-operation. The sampling height coincided with the height in which the stationary inlet was placed. The distance from the point-of-operation was carefully considered so that measurements were not impeding on a given activity. One monitoring session was performed per day to ensure the conditions were controlled. The HVAC system was disabled when the windows were opened and enabled when windows were closed. We selected areas of the rooms where the supply inlets and exhaust outlets were furthest away from the measurements. Additionally, an outdoor dataset was collected, including seven replicates averaging 66 min of sampling time (ranging from 47 to 112 min).

The raw data files from the InstaScope consisted of size-resolved and fluorescence-categorized particle number counts based on the characteristics of each detected particle with millisecond resolution. Each data point collected was classified with respect to particle type, type of activity, and experimental replication. Python software was utilized to obtain total particle counts and concentrations for each particle type, including fluorescent and non-fluorescent particles from each experiment. The obtained concentrations were an average for each replicate experiment for the entire sampling session (28–224 min). Thus, there were 5 data points for each activity. The response variable was a normalized concentration calculated as a concentration ratio (C_Ratio_) for each data point as follows:C_Ratio_ = (C_Activity_)/C_Background_
(1)
where: C is Concentration. C_Activity_ is the concentration of each particle type during an activity. C_Background_ is the average concentration of each particle type during background.

The data processing and analyses were conducted on R Statistical Software (v4.1.1). The data were cross-classified according to two factors. One factor was particle type with five levels (pollen, fungi, bacteria, non-fluorescent, and fluorescent particles), while the other was the type of activity with seven levels (bedsheet changing, cooking, dusting, showering, sleeping, vacuuming, and walking). The concentration ratio was then log-transformed (natural log) to meet the normality assumption required for the validity of statistical analyses. The raw particle count over time was then graphed to show real-time particle generation for specified activities selected based upon the results of the statistical tests.

The first line of analysis tested for differences between activities as well as particle types via the analysis of variance (ANOVA) method. The normality of the data was tested to verify the applicability of the method, using a one percent level of significance (α ≥ 0.01). In order to obtain a clearer picture of comparisons, we pursued Tukey’s Honest Significant Difference method on each factor. The second line of analysis was to compare activities by each of the five particle types. Therefore, we applied the ANOVA method for one-way classified data to compare differences between activities for each type of particle. If the homogeneity of activity means for a given particle type was rejected, we made pairwise comparisons of activities via Tukey’s Honest Significant Difference method enhanced by the Compact Letter Display (CLD) graph. The third line of analysis was to check for differences between the particle types for each activity. Finally, we pursued Tukey’s Honest Significant Difference method enhanced by CLD graphs for the cases when the homogeneity hypothesis was rejected.

## 3. Results

For each particle type, the concentrations were averaged over all activities (excluding background and outdoor experiments) and are shown in Table 1 along with standard deviations. Background and outdoor concentrations and standard deviations are shown separately. Higher concentrations of pollen and fungi were measured within background, outdoor, and activity experiments compared to bacteria. The concentrations of fluorescent particles were higher than non-fluorescent particles in the background and outdoor experiments and during the activities. 

We confirmed the normality assumption (*p* = 0.02) to justify the applicability of the two-way ANOVA method. The homogeneity of population means of the natural log-transformed concentration ratios by activity as well as by particle type was rejected (*p* < 0.001 in each case) (Table 2), indicating that there were significant differences in concentration ratios between activities and between particle types. This prompted us to pursue pairwise comparisons between activities as well as particle types. 

Figure 2 shows the CLD graphs summarizing the results from pairwise comparisons of all activities adjusted for all particle types (Figure 2A) as well as all particle types adjusted for all activities (Figure 2B) according to Tukey’s HSD method. We observed that the log-transformed concentration ratios for the showering experiments were significantly different (higher) from all other activities, as seen in Figure 2A indicated by the letter “d”. Showering resulted in a median log ratio of 2.67, corresponding to 14.43 in the original scale. Sleeping experiments appeared to result in significantly lower concentration ratios from all other activities as indicated by the letter “a” in Figure 2A, with a median log ratio of −0.90, corresponding to 0.41 in the original scale. Pairwise analysis of all particle types resulted in three distinct clusters (indicated by the letters “a”, “b”, and “c”), as seen in Figure 2B. No two mean concentration ratios of bacterial, non-fluorescent, and fluorescent particles were significantly different. Lastly, the two mean concentration ratios of pollen and fungal particles were not significantly different.

Following the second line of analysis, the results of the ANOVA method are reported in Table 3. The table provides *p*-values associated with the null hypothesis of homogeneity of activity means for each particle type. The homogeneity hypothesis was rejected for all particle types. Tukey’s HSD method was applied to examine pairwise differences between activities, supplemented by the CLD graph (Figure 3).

The box plots of bacterial concentration ratios under the seven activities aligned into two clusters, identified by the letters “a” and “b” in Figure 3A. Bedsheet changing and walking appeared to produce the highest and lowest median concentration ratios of 1.47 (corresponding to 4.35 in the original scale) and −1.01 (corresponding to 0.36 in the original scale) for bacteria, respectively. In Figure 3B, the box plots of the concentration ratios of pollen aligned into four clusters, identified by the letters “a”, “b”, “c”, and “d”. Walking and sleeping appeared to produce the highest and lowest median concentration ratios of 4.26 (corresponding to 70.81 in the original scale) and −0.98 (corresponding to 0.38 in the original scale) for pollen, respectively. In Figure 3C, the box plots of concentration ratios of fungi aligned into three clusters, identified by the letters “a”, “b”, and “c”. Showering and sleeping appeared to produce the highest and lowest median concentration ratios of 2.88 (corresponding to 17.81 in the original scale) and −0.44 (corresponding to 0.64 in the original scale) for fungi, respectively. The differences in the concentration ratios of non-fluorescent particles (Figure 3D) and fluorescent particles (Figure 3E) exhibited similar differences between activities as observed for fungi (Figure 3C).

The homogeneity of the mean concentration ratios of all particle types was tested by the ANOVA method for each activity (Table 4). In five activities, the hypothesis of homogeneity of population means of particle types was not rejected (*p* > 0.05). In two activities (showering and walking), the hypothesis was rejected (*p* ≤ 0.05), indicating that there were significant differences in concentration ratios of particle types during showering and during walking. Therefore, we further compared the mean concentration ratios between particle types for showering and walking activities using Tukey’s HSD method enhanced by CLD graphs, as shown in Figure 4.

The concentration ratios of bacteria for showering appeared to be significantly different from the other concentration ratios, as identified by the letter “a” (Figure 4A). Furthermore, no two concentration ratios of pollen, fungi, and non-fluorescent and fluorescent particles were significantly different within the showering experiments. Fungi and bacteria produced the highest and lowest median concentration ratios of 2.88 (corresponding to 17.81 in the original scale) and 0.64 (corresponding to 1.90 in the original scale) for showering, respectively. Within the walking experiments, we observed that the concentration ratios of pollen were significantly different from each of the other particle types, as identified by the letter “d” (Figure 4B). Pollen and bacteria produced the highest and lowest median concentration ratios of 3.96 (corresponding to 52.46 in the original scale) and −1.01 (corresponding to 0.36 in the original scale) for walking, respectively.

The real-time data are shown for walking and showering experiments as examples (Figure 5). In both activities, the concentrations of all the particle types appeared to follow a similar pattern (Figure 5). Figure 5A shows two peaks: one in the beginning and one at approximately 25 min. In Figure 5B, there is a steady increase with a distinct peak at 20 min. The exception from these was the concentration of bacteria as they appeared to be low and did not show any specific trends.

## 4. Discussion

To our knowledge, no prior study has been conducted that captures home care activities such as bedsheet changing, cooking, dusting, showering, sleeping, vacuuming, and walking and compares them to particle type including bacteria, pollen, fungi, fluorescent, and non-fluorescent particles. We found that all the studied activities, except sleeping, increased the concentrations of all studied particle types compared to background. Home healthcare workers assist in performing often-occurring everyday activities such as vacuuming, walking, and showering. When testing for differences in particle types between activities, showering and walking had significant differences between particle types.

Our vacuuming experiments produced median concentration ratios of 0.94, 0.24, 0.76, and 0.53 (2.56, 1.27, 2.14, and 1.70 in the original scale, respectively) for pollen, bacteria, fungi, and fluorescent particles, respectively. Particles emitted from vacuuming activities may vary depending on the characteristics of the home, occupancy, and type of vacuum. In addition to being reaerosolized from surfaces, bacteria and fungi may remain viable within the dust bag for up to two months and be resuspended into the air during vacuuming [50]. In a study published by Knibbs et al., researchers utilized closed-face cassettes (37 mm) loaded with polytetrafluoroethylene filters to measure bacteria during vacuuming activities [34]. Mean bacterial emission rates ranged from 0 to 6.4 × 10^5^ bacteria min^−1^ and 3.0 × 10^3^ to 7.4 × 10^5^ bacteria min^−1^ for cold and warm vacuums, respectively [34]. Our results indicated that pollen, bacteria, fungi, and fluorescent particle types were 17, 2.25, 6, and 4 times greater on average than background measurements, respectively. Particle generation from vacuuming could have been attributed to factors such as occupancy, type of vacuum, and resuspension. Similar experiments conducted by Corsi et al. studied particle resuspension from vacuuming and concluded that the act of vacuuming could be a significant source of PM_10_ [51]. The authors also advised that particles may remain airborne for several hours after vacuuming. Many studies investigating the effects of vacuuming also found increased levels of fungi, with concentrations decreasing within 60 to 90 min [52].

We measured median concentration ratios of 1.77 and 0.85 (5.87 and 2.34 in the original scale, respectively) for fungi during bedsheet changing and dusting experiments, respectively. Jürgensen et al. investigated fungal concentrations during bed making and sweeping, similar to dusting [52]. Their study showed that levels of culturable airborne fungi increased by a factor of 3.2 during bed making at different points of the activity. Furthermore, studies have shown that fungal particles did not reach background levels within 30 to 60 min after the activity was completed [53]. Similar results were seen in a study investigating a sweeping activity (increase in fungal spore concentration by a factor of 7.3). After the activity was completed, the fungal levels began to decrease over time but were unable to reach background levels within 30–60 min [52].

Our results show that walking was associated with higher concentration ratios of particles (Table 4 and Figure 4B). Walking experiments produced a median concentration ratio of 3.96, 1.02, and −1.01 (52.46, 2.77, and 0.36 in the original scale, respectively) for pollen, fungi, and bacteria, respectively. Although the effects of walking on fungi have not been well characterized in residential environments, studies exist which investigate walking in public environments. During regular foot traffic, levels of fungi were significantly increased [54]. Walking has also resulted in the resuspension of fungal spores in an experimental room fitted with nylon carpet and fungal spores [55]. One study evaluated fluorescent particles associated with skin shedding while walking in a chamber [56]. The mean emission rates of total supermicron fluorescent particles from walking ranged from 6.8 to 7.5 million particles per person-hour [56]. Our experiments showed that walking produced slightly higher biological particles, which could be attributed to the frictional interaction between clothing and skin, which has been shown to be a source of bioaerosol emissions from humans [56]. Our walking experiments were consistent also with another chamber study that showed that walking enhanced resuspension of particles by five to six times compared to sitting [30]. Our study measured both fluorescent and non-fluorescent median particle concentration ratios of 1.55 and 1.66 (4.71 and 5.26 in the original scale, respectively) during cooking experiments. Tian et al. [37] concluded that indoor concentrations of fluorescent particles were strongly influenced by human occupancy and activities such as cooking. Cooking activities tested during their study increased indoor total and fluorescent particle concentrations by two orders of magnitude above background levels measured during unoccupancy. Researchers also found that splashing of cooking material/ingredients is a source of indoor fluorescent particles [37]. Similarly, Kanaani et al. [57] found that aerosolized cooking oil produced fluorescent particles. For our study, potential sources for biological particles during cooking and showering activities could have been from direct shedding from occupants, aerosolized biological particles from water and food processing (during cooking), and resuspension from clothing [30,33,52]. 

Studies have documented that bacterial populations are associated with showering (e.g., showerheads, curtains, and water) [58,59,60]. Our showering experiments resulted in higher concentrations of bioaerosol, which corroborate previous findings. The showering experiments included the act of showering, which included removing clothes, washing the body, then drying the body. The higher levels of bioaerosols from showering within our experiments may be attributed to such factors. One study investigated domestic shower hoses and their fungal and bacterial populations and found aquatic and environmental bacteria within samples [61]. The work conducted by Moat et al. verified that domestic showering could result in exposure to bioaerosols such as bacteria and fungi [61].

Our current study showed that sleeping activity yielded the fewest particles measured, which could be attributed to inactivity during sleeping. These data are consistent with other studies focusing on indoor bioaerosol concentrations, such that concentrations are strongly correlated with household occupancy and activity [14]. During our sleeping experiments, when occupants did not engage in activities that would cause particles to be generated or resuspended, the concentration of particles was below background averages. Similarly, Chen and Hildemann have reported increased residential bioaerosol concentrations during daytime hours where occupants are engaging with the environment via other activities [23]. However, pillows and mattresses serve as reservoirs for settled dust consisting of bacteria, fungi, and allergens, which can be resuspended upon movement during sleep. Various factors could have influenced the low concentration of bioaerosol during sleeping, such as occupants not tossing and turning during sleep [39]. In addition to bedroom dynamics, the type of mattresses and bedding materials have been documented to influence levels of biological matter. A study by Bemt et al. reported innerspring mattresses having higher levels of biological matter versus latex and polyester mattresses [62]. Fungal species have also been found within pillow foam and mattresses as they serve as an ideal medium for culturing induced by sweating, moisture, and elevated temperatures [60,61]. Bacterial phyla and genera have also been documented in sleeping environments to be mainly of human origins (shedding, genital, oral, fecal, etc.) [33,63].

The comparison between our concentration values to other studies is limited since we utilized a novel particle characterization technique. Li et al. [27] used the WIBS in a residence in Singapore and reported fluorescent particle concentrations of 700 and 1400 particles/L for indoor and outdoor environments, respectively. A similar study by Tian et al. [14] used laser-induced fluorescence (Ultraviolet Aerosol Particle Sizer) and reported that daily average concentrations of fluorescent particles in the size range of 1–10 µm were 40 particles/L during summer and 29 particles/L during winter in Californian residences. Lastly, Nieto-Caballero et al. [42] assessed mold-like particles using WIBS in Colorado schools before and after renovation. They reported median concentrations of 7 and 3 particles/L for mold-like particles [42]. Our average concentrations for fluorescent particles (average ranging from 578 to 2240 particles/L) were comparable with the study of Li et al., who, similar to us, reported non-size-selective concentrations. Tian et al. [14] reported lower concentrations compared to our study, most likely because they used a different instrument and reported results within a selected size range. Our average fungal concentrations in outdoor air (7 particles/L) were similar to those reported by Nieto-Caballero for schools [42]. However, our indoor concentrations were higher: averages of 22 particles/L for background and 139 particles/L during activities. This can be explained by more diverse sources for fungal particles in homes versus schools.

Among the specific type of bioaerosols, the concentration of pollen was the highest, and the concentration of bacteria was the lowest. We were not able to find other studies that have reported pollen and bacteria concentrations in indoor environments based on measurements with laser-induced fluorescence. It should be noted that our study likely observed pollen fragments in addition to intact pollen grains since pollen fractionates in humidity gradients as they age. For instance, a pollen particle with an intact diameter of around 30 μm may undergo osmotic shock and rupture, which releases hundreds of pollen fragments ranging in size from 0.60 to 2.5 μm [64,65,66]. Similarly, our study may have also observed fungal fragments from contaminated surfaces. For instance, up to 56 percent of total fungal particle counts in field samples have been shown to be hyphal fragments based on microscopic sample analysis [66,67].

Although our study was limited to one studied residence, the experiments produced results similar to published data, which shows a promising future for the detection of bioaerosols. A significant strength of our experiments was the utilization of the InstaScope, which produced real-time results, thus eliminating the requirement, equipment, time, and costs associated with traditional laboratory analysis. The InstaScope was the first fluorescence-based direct-reading instrument that referenced an optical library of signatures for characterizing bioaerosols without speciation. The InstaScope’s optical library has been validated through laboratory studies that have recognized precise fluorescence properties, coupled with optical diameters, which can then discern between different types of bioaerosol types (pollen, fungi, and bacteria) [45]. The mobility of the InstaScope allowed us to monitor activities of daily living and record bioaerosol measurements.

## 5. Conclusions

Our results provide baseline data regarding bioaerosol concentrations during activities that are commonly encountered by home healthcare workers. We concluded that showering and sleeping activities produced the highest and lowest increases in median concentration ratios, respectively. Analysis by individual particle types showed that bedsheet changing produced the highest increase in the median concentration of bacteria, showering produced the highest increase in the median concentration of fungi, fluorescent, and non-fluorescent particles, and walking produced the highest increase in the median concentration of pollen. Based on our results, this study will allow researchers to further expand on bioaerosol exposures within home healthcare as they investigate different types of residences, variation in occupancy, and other conditions. We recommend that future experiments be conducted within real patients’ homes during HHCWs’ work shifts. Instruments such as the InstaScope will allow practitioners within home healthcare to determine the conditions of a given residence, which will pave future preventative measures to protect workers within an unregulated environment. We further conclude that utilizing direct-reading methods may save time and effort in bioaerosol-exposure assessment.

## Figures and Tables

**Figure 1 ijerph-19-03613-f001:**
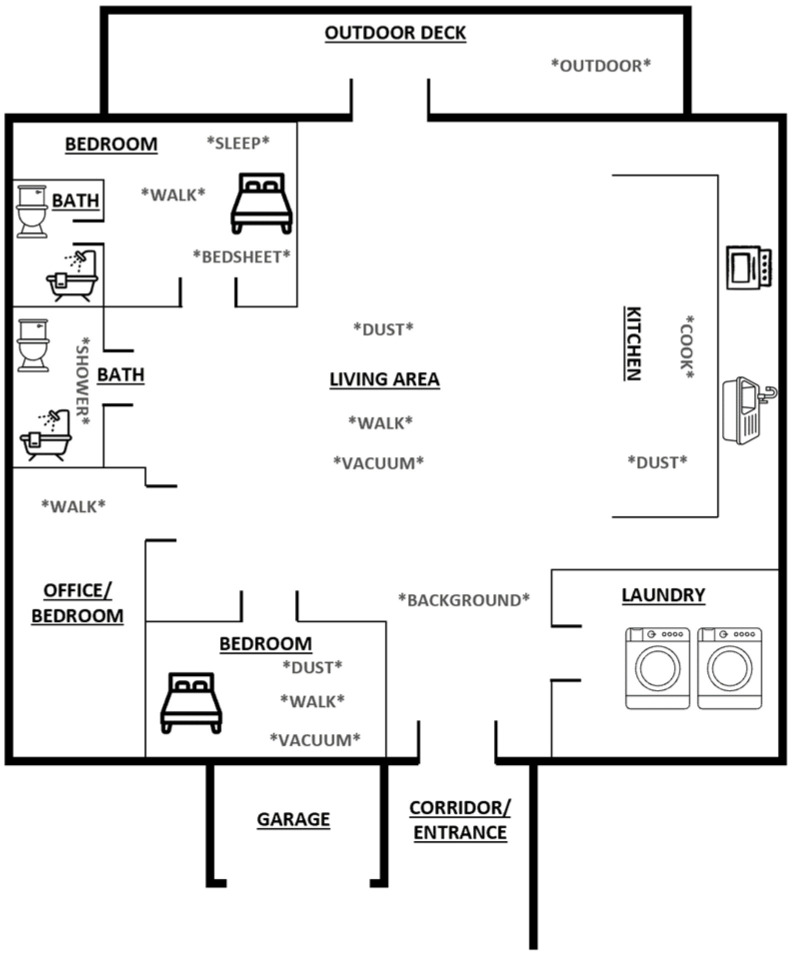
The studied residence included three bedrooms and two bathrooms. The tested bedroom, bathroom, kitchen/dining room, and living room were approximately 3, 20, 40, and 24 m^2^, respectively. The areas where measurements were taken are indicated by “*”.

**Figure 2 ijerph-19-03613-f002:**
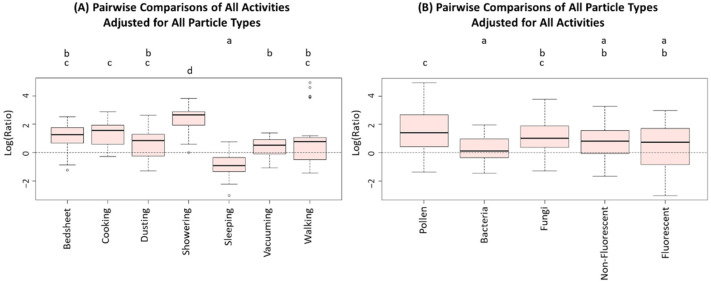
The Compact Letter Display (CLD) graphs portraying pairwise differences between (**A**) Activities, and (**B**) Particle Types. The data behind the comparisons are the natural-log transformed concentration ratios. In each CLD graph, the five-point box plot shows the minimum, first quartile, median, third quartile, and maximum of 25 data points for each activity (except sleeping, *n* = 20), and 34 data points for each particle type. The dotted line in each CLD graph shows log-ratio of zero (or equivalent anti-log ratio of one). Any two treatments receiving the same letter at the top of the graphs are not significantly different as per Tukey’s HSD method.

**Figure 3 ijerph-19-03613-f003:**
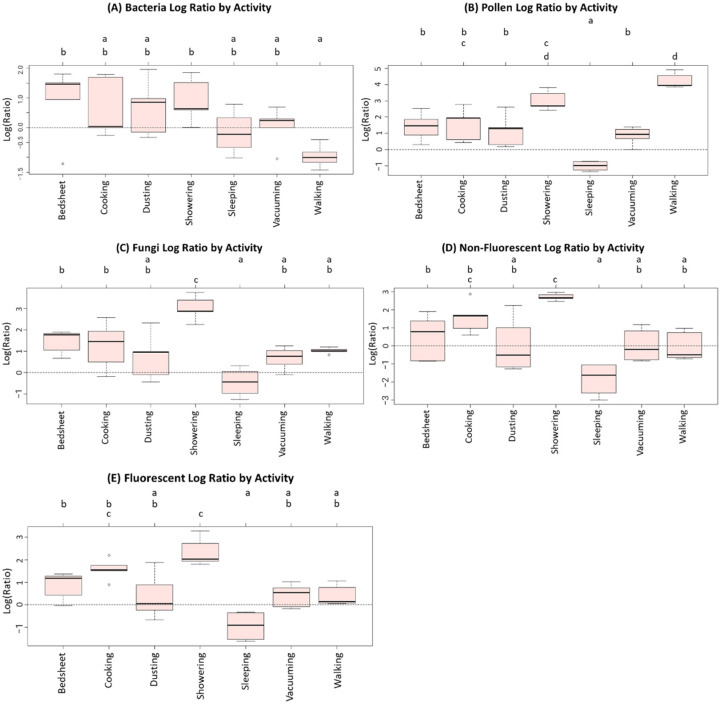
The Compact Letter Display (CLD) graphs portraying pairwise differences between all activities by each particle type. The data behind the comparisons are the natural-log transformed concentration ratios. The graphs are indexed by five particle types (**A** = Bacteria, **B** = Pollen, **C** = Fungi, **D** = Non-Fluorescent, **E** = Fluorescent). In each CLD graph, the five-point box plot shows the minimum, first quartile, median, third quartile, and maximum of five data points for each activity (except sleeping, *n* = 4). The dotted line in each CLD graph shows log-ratio of zero (or equivalent anti-log ratio of one). Any two treatments receiving the same letter at the top of the graphs are not significantly different as per Tukey’s HSD method.

**Figure 4 ijerph-19-03613-f004:**
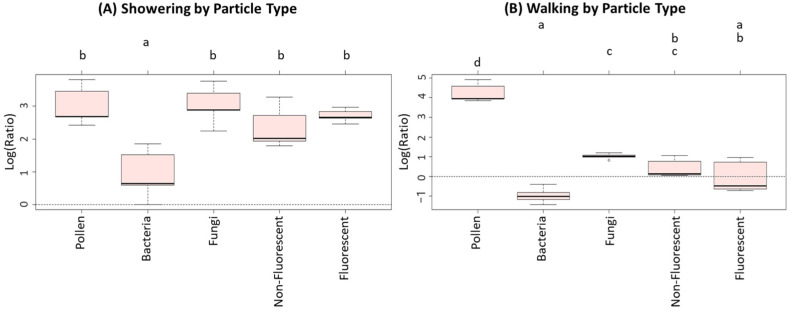
The Compact Letter Display (CLD) graphs portraying pairwise differences between showering and walking by each particle type. Log-transformed concentration ratios were used to check on pairwise differences among the type of particles for showering and walking (**A** = Showering and **B** = Walking) by applying Tukey’s HSD method, whose results were summarized by the CLD graphs. In each CLD graph, the five-point summary was portrayed as a box plot (minimum, first quartile, median, third quartile, maximum) of five data points for each particle type within the activity. The dotted line in each CLD graph was drawn at log-ratio zero or, equivalently, ratio one. Any two treatments receiving the same letter at the top of the graphs are not significantly different as per Tukey’s HSD method.

**Figure 5 ijerph-19-03613-f005:**
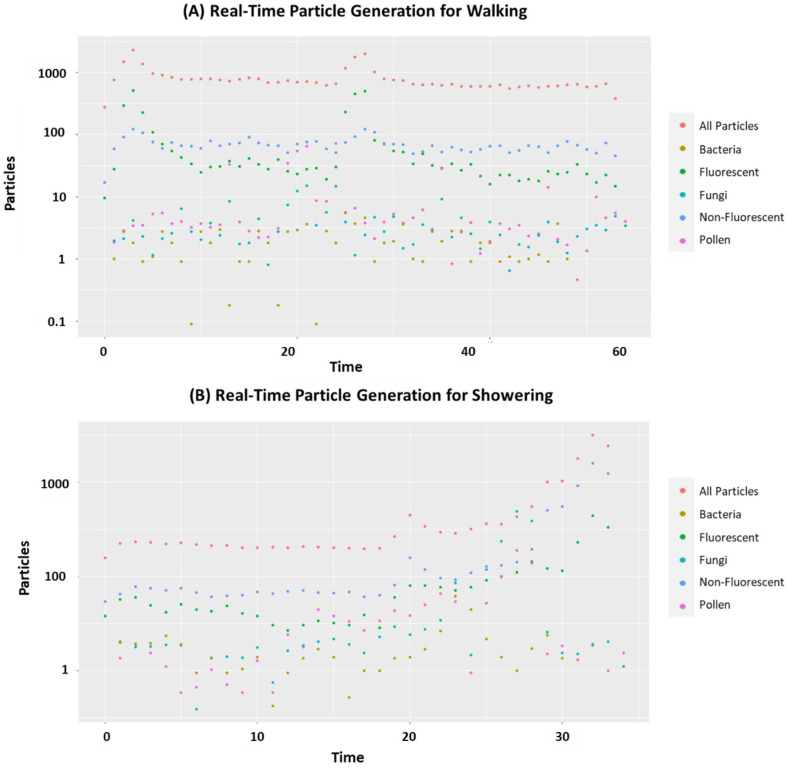
Real-time particle concentrations on log scale over time for one replicate of both (**A**) walking and (**B**) showering activities.

**Table 1 ijerph-19-03613-t001:** Average concentrations (particles/liter) and standard deviations for particle types within all activities, in background and outdoors.

Particle Type	Concentration During Activities (*n* = 34)	Background Concentrations (*n* = 5)	Outdoor Concentrations (*n* = 7)
Pollen	242 ± 405	14 ± 5	27 ± 21
Bacteria	9 ± 9	4 ± 2	0.7 ± 0.4
Fungi	139 ± 198	22 ± 6	7 ± 4
Fluorescent	2240 ± 2899	578 ± 255	1213 ± 994
Non-Fluorescent	621 ± 799	140 ± 120	178 ± 85

**Table 2 ijerph-19-03613-t002:** Results of two-way ANOVA testing homogeneity of means of concentration ratios by activity and particle type after the log-transformation of the data ^1^.

	Degrees of Freedom	Sum Sq ^2^	Mean Sq	F-Value	*p*-Value
Activity	6	133.77	22.295	22.02	<0.001
Particle Type	4	40.82	10.206	10.08	<0.001
Residuals	159	160.95	1.012		

^1^ Concentration data obtained from the InstaScope. ^2^ Square is abbreviated as Sq.

**Table 3 ijerph-19-03613-t003:** Results of testing the homogeneity of means of concentration ratios by particle type with all activities included.

Particle Type	*p*-Value
Bacteria	0.015 *
Pollen	<0.001 *
Fungi	<0.001 *
Non-Fluorescent	<0.001 *
Fluorescent	<0.001 *

* Denotes significance.

**Table 4 ijerph-19-03613-t004:** Results of testing homogeneity of means of particle concentration ratios by each activity with all particle types included.

Activity	*p*-Value
Bedsheet Changing	0.47
Cooking	0.463
Dusting	0.632
Showering	<0.001 *
Sleeping	0.061
Vacuuming	0.219
Walking	<0.001 *

* Denotes significance.

## Data Availability

De-identified data are available upon request and approval by the PI.

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
