# Peer review of "Direct-Read Fluorescence-Based Measurements of Bioaerosol Exposure in Home Healthcare"

_ijerph, 2022, doi:10.3390/ijerph19063613_

Round 1

Reviewer 1 Report

Review

Material and method

Please explain which and technical characteristics the tools you use for the measurements.

Results

Table 1 line 207 please insert the data source

``Similar experiments conducted by Corsi et al. studied parti…`` please complete correct citation within the text- line 334

Conclusion section should be improved.

Reviewer 2 Report

The authors presented a study characterizing the concentrations of three types of bioaerosols based on a novel fluorescence-based direct reading instrument. This study is interesting and the conclusion is well supproted by the data. I recommended the manuscript to be accecpted upon the following comments being considered.

1) In the Section "2. Materials and Methods", it is suggested to use a schematic diagram to describe the tested residence. The words are somehow boring.

2) What do you mean by "each activity was replicated five times"? So the data was averaged to get the final data points?

3) Please explain the acronym such as Sq, F-Value, P-Value.

4) Why was the data log-transformed?

Reviewer 3 Report

The authors in their article took up an interesting research problem and presented the obtained valuable results. However, the article will gain value if my following remarks and comments are taken into account: 

  • In the Introduction, the authors devoted a lot of attention to the description of "home health workers", but it has absolutely no impact on the planned research cycle and conducted measurements, and the discussion of the results. In lines 114-115 there is information only about 3 persons and a cat living in the place of research. Were any of them a "home health worker"? They might as well be miners, or sailors. In line 57 the authors define bioaerosols as ".. parts of biological oigin .." This is not true. Bioaerosols are always a two-phase system consisting of gas (air). 
  • In lines 67-69 the authors list the sources of organic particles in indoor bioaerosols. What is missing here are those derived from indoor animals and plants. 
  • The "materials and methods" section does not describe exactly how the measurements were performed. Is their conduct based on relevant standards? At what height and in what parts of the rooms were the measurements carried out? Why was it not described how and how often the air conditioning system was serviced? Why were the air conditioner filters not tested? Why was the description not included in the description of the possible impact of air conditioners on indoor air movement and the displacement of bioaerosols? 

Reviewer 4 Report

In my opinion, the paper has major shortcomings and needs improvements. According to Huffman et al., 2019 (https://doi.org/10.1080/02786826.2019.1664724), the Instascope (DetectionTek; Boulder, Colorado, USA) is used commercially to monitor mold and fungal spores inside homes, lacks asymmetry measurement and is primarily marketed to non-scientists. In my opinion, the instrument is not suitable to measure with accuracy the concentration of bacterial and fungal bioaerosols. The concentration of bacterial bioaerosols is usually express in colony-forming unit (CFU). In this paper the concentration of bacterial bioaerosols is express as a ratio.

I suggest to collect the bioaerosols using a standardized method (petri dishes plates containing sterile culture medium). After a period of incubation of the samples, the identification of bacterial and fungal bioaerosols should be performed. The concentration of bioaerosols should be presented in a figure in the results section along with errors bars standard deviations of three replicates. A comparison of the concentration levels of bioaerosols resulted from the investigated activities (bedsheet changing, cooking, dusting, showering, sleeping, vacuuming, and walking) should be made in the results section. The variations of the bacterial and fungal concentration resulted from the investigated activities should be discussed in details. The findings should be discussed and compared with other previous.

Round 2

Reviewer 4 Report

The authors have well addressed all my comments. The overall quality of the paper was improved.